# The effectiveness of smoking cessation, alcohol reduction, diet and physical activity interventions in changing behaviours during pregnancy: A systematic review of systematic reviews

**Nicola Heslehurst**[1,2]*, **Louise Hayes**[1,2], **Daniel Jones**[3], **James Newham**[4], **Joan Olajide**[2,3], **Louise McLeman**[1], **Catherine McParlin**[2,5], **Caroline de Brun**[3], **Liane Azevedo**[2,6]

1 Population Health Sciences Institute, Newcastle University, Newcastle upon Tyne, United Kingdom, 2 Fuse, The Centre for Translational Research in Public Health, Durham, United Kingdom, 3 School of Health and Life Sciences, Teesside University, Middlesbrough, United Kingdom, 4 School of Life Course Sciences, King's College London, London, United Kingdom, 5 Department of Nursing Midwifery and Health, Northumbria University, Newcastle upon Tyne, United Kingdom, 6 School of Human and Health Sciences, University of Huddersfield, Huddersfield, United Kingdom

* nicola.heslehurst@ncl.ac.uk

## Abstract

### Background

Pregnancy is a teachable moment for behaviour change. Multiple guidelines target pregnant women for behavioural intervention. This systematic review of systematic reviews reports the effectiveness of interventions delivered during pregnancy on changing women's behaviour across multiple behavioural domains.

### Methods

Fourteen databases were searched for systematic reviews published from 2008, reporting interventions delivered during pregnancy targeting smoking, alcohol, diet or physical activity as outcomes. Data on behaviour change related to these behaviours are reported here. Quality was assessed using the JBI critical appraisal tool for umbrella reviews. Consistency in intervention effectiveness and gaps in the evidence-base are described.

### Results

Searches identified 24,388 results; 109 were systematic reviews of behaviour change interventions delivered in pregnancy, and 36 reported behavioural outcomes. All smoking and alcohol reviews identified reported maternal behaviours as outcomes (n = 16 and 4 respectively), whereas only 16 out of 89 diet and/or physical activity reviews reported these behaviours. Most reviews were high quality (67%) and interventions were predominantly set in high-income countries. Overall, there was consistent evidence for improving healthy diet behaviours related to increasing fruit and vegetable consumption and decreasing

**Data Availability Statement:** All relevant data are within the paper and its Supporting Information files.

**Funding:** NH and LA received some funding towards this research from Fuse, the Centre for Translational Research in Public Health (www.fuse.ac.uk). The funding was used to pay for an information scientist (CB) to develop and carry out the search strategy. No further funding was received for this work. NH, LA, LH, CM, and JO are members of Fuse, the Centre for Translational Research in Public Health. Fuse is a UK Clinical Research Collaboration (UKCRC) Public Health Research Centre of Excellence. Funding for Fuse from the British Heart Foundation, Cancer Research UK, National Institute of Health Research, Economic and Social Research Council, Medical Research Council, Health and Social Care Research and Development Office, Northern Ireland, National Institute for Social Care and Health Research (Welsh Assembly Government) and the Wellcome Trust, under the auspices of the UKCRC, is gratefully acknowledged. The views expressed in this paper do not necessarily represent those of the funders or UKCRC. The funders had no role in study design, data collection and analysis, decision to publish, or preparation of the manuscript.

**Competing interests:** The authors have declared that no competing interests exist.

carbohydrate intake, and fairly consistent evidence for increase in some measures of physical activity (METs and $VO_2$ max) and for reductions in fat intake and smoking during pregnancy. There was a lack of consistent evidence across reviews reporting energy, protein, fibre, or micronutrient intakes; smoking cessation, abstinence or relapse; any alcohol behaviours.

## Conclusions

The most consistent review evidence is for interventions improving dietary behaviours during pregnancy compared with other behaviours, although the majority of diet reviews prioritised reporting health-related outcomes over behavioural outcomes. Heterogeneity between reported behaviour outcomes limits ability to pool data in meta-analysis and more consistent reporting is needed. Limited data are available for alcohol interventions in pregnancy or interventions in low- or middle-income-countries, which are priority areas for future research.

## Introduction

There are multiple complex public health priorities that healthcare and public service providers internationally are tackling, particularly relating to smoking, alcohol, poor nutrition and sub-optimal levels of physical activity [1, 2]. Strategies for public health interventions include environmental change such as improving home, work and school environments, and smoking bans in public spaces; fiscal, policy and regulation changes such as levies on unhealthy foods (e.g. sugar tax and minimum alcohol pricing); industry change such as front of package health warnings on tobacco, food labelling and restrictions on marketing and advertising; and strategies to promote individual behaviour change such as health professional support and counselling [1, 2]. Specifically relevant to individual behaviour change is the concept of teachable moments, defined as *"naturally occurring events thought to motivate individuals to spontaneously adopt risk-reducing health behaviours"* [3]. Pregnancy is argued to be a prime teachable moment, with women undergoing a life transition whilst in frequent contact with healthcare professionals and service providers [4]. For many women and their families, this may be their first encounter with healthcare services. In the UK, the National Institute for Health and Care Excellence (NICE) guidance on behaviour change emphasises the importance of intervening at key life stages or times, including pregnancy [5]. An underlying assumption is that behaviours change due to women prioritising fetal health and also responding to social norms on the acceptability of certain behaviours in pregnancy. A major question is whether or not there are similarities or differences across health behaviours in women's propensity for positive change, and whether there are shared lessons that can be learned for some behaviours that could be translated to others.

Key public health priorities that are strongly associated with health and social inequalities, and are particularly relevant to pregnancy due to potential immediate risk to the mother and/or fetus, include diet and physical activity behaviours (linked to maternal obesity and gestational weight gain), smoking and alcohol use. These behaviours form the focus of this systematic review of systematic reviews. Having a healthy diet and being physically active during pregnancy can have benefits for the mother and her baby, including improved cardiorespiratory fitness, reduction in gestational diabetes and caesarean section [6–9]. However, physical

activity levels tend to decline throughout the course of pregnancy and dietary quality is often suboptimal [10, 11]. Poor maternal diet and physical activity behaviours can be associated with both pre-pregnancy obesity and excessive gestational weight gain, which further increases risks to mother and fetus (including gestational diabetes, pre-eclampsia, macrosomia, caesarean section and pre-term birth), and have implications for healthcare resources [12–14]. Prevalence of smoking among pregnant women in high income countries has declined from 20 to 35% in the 1980s to between 10% and 20% in the early 2000s, with a further decline to below 10% by 2010 [15]. However, a different picture is evident amongst socially disadvantaged women and among women in low income countries, where smoking prevalence has not decreased and in some places continues to increase [16, 17]. Health risks of smoking in pregnancy include spontaneous abortions, ectopic pregnancies, placenta previa and babies being small for gestational age [18, 19]. Prevalence of alcohol use in pregnancy varies by country, it is lowest in the World Health Organisation (WHO) Eastern Mediterranean Region (EMR) (0.2%, 95% CI: 0.1–0.9) and highest in the WHO European region (25.2%, (21.6–29.6). Prenatal alcohol exposure is associated with preterm birth, low birth weight and Fetal Alcohol Spectrum Disorders [20–22].

In light of the potential for harm relating to these behaviours during pregnancy, and the concurrent potential for public health gain through intervention, there are national and international guidelines for diet and physical activity behaviours (including weight management) and smoking cessation [23–25]. Guidance on alcohol consumption is variable, with advice ranging from abstinence to light consumption (usually defined as no more than 1 to 2 units, once or twice a week) [26–28].

The extent of adoption of behaviour change among pregnant women is unclear [29]. Pregnant women may make changes to a range of health behaviours under pressure from societal norms [30]. Examples of changes in behaviour include 'spontaneous quitters' among women who smoke, though many do not continue to abstain beyond pregnancy, arguably due to a newly defined self-concept and perceived social stigmatisation [3]. Weight management research shows some women make positive dietary changes due to prioritising the health of their unborn baby [31], whereas their perceptions of physical activity risk during pregnancy might influence this behaviour [32]. Existing evidence suggests that many barriers to being physically active during pregnancy exist, including lack of consistent advice, lack of societal support, physical symptoms of pregnancy and lack of opportunity [33]. Factors that motivate women to be active during pregnancy include weight control and potential for an easier labour and delivery [33]. Some women may perceive drinking small amounts of alcohol during pregnancy as important for their social lives which could also influence their behaviour [34].

There is clear emphasis on pregnant women as a target for intervention across different behavioural domains [35]. However, interventions in pregnancy, and subsequent evidence synthesis of interventions, tend to be carried out in silos with no attempt to synthesise across behaviours despite the overlap in target population. Further, when considering translation of effective interventions into routine practice, the same health professionals are responsible for delivering multiple interventions simultaneously [36]. The increasing need for multiple interventions, and development of a plethora of referral systems, pathways and guidelines, can ultimately present a significant burden to women, healthcare professionals and services trying to manage complex pregnancies [37]. Synthesising evidence on interventions focusing on different behavioural domains (smoking, alcohol consumption, diet and physical activity) would enhance our understanding of the effectiveness of interventions which target behaviour change during pregnancy and could identify shared mechanisms of what types of intervention are effective and for whom, as well as identification of similarities or differences across different target behaviours. Such information could help to inform an interdisciplinary approach to

public health around pregnancy and guide the development and delivery of cost-effective interventions with the potential to impact on short-term and long-term health outcomes for women and children. These outcomes include improved maternal weight management, reduced risk of gestational diabetes, and hypertension, lower rates of premature delivery and caesarean section and of low and high birthweight infants.

This paper is the first to be reported from a wider programme of systematic reviews of systematic reviews exploring diet, physical activity, smoking and alcohol behavioural interventions delivered during pregnancy. The wider programme aims to 1) examine the effectiveness of interventions on changing behaviour in pregnant women, 2) examine the effectiveness of interventions on improving pregnancy outcomes, and 3) explore any shared behavioural techniques or content of interventions that may be associated with effectiveness across these behavioural domains. This paper reports the systematic review of systematic reviews addressing aim 1.

## Materials and methods

This systematic review of systematic reviews provides an overview of the existing evidence base. It compares findings of previous systematic reviews, identifies research gaps and provides direction for future research, specifically relating to intervention effectiveness across behavioural domains. We used the Joanna Briggs Institute (JBI) methodology for umbrella reviews [38] and the PRISMA reporting guidelines and checklist (S1 PRISMA Checklist) [39]. The protocol for this systematic review has been published [40] and was registered on the PROSPERO database (CRD42016046302).

### Identification of studies

A comprehensive search of fourteen bibliographic databases was originally conducted in May 2016, and updated in March 2018 and November 2019. The databases searched were: Joanna Briggs Institute Database of Systematic Reviews and Implementation Reports, JBI COnNECT +, Cochrane Database of Systematic Reviews, Cochrane Database of Abstracts of Reviews of Effectiveness (DARE), Cochrane Health Technology Assessments, Cochrane Economic Evaluations, PROSPERO, Epistemonikos, Medline, CINAHL, AMED, ASSIA, LILACS, and Social Care Online. Grey literature was also included in the original search. The following databases were searched: Google Scholar, NICE Evidence search, Open Grey, The Grey literature report, National Institute for Health Research (NIHR) Journals Library, Health Technology Assessment Database, Ovid Health Management Information Centre Database, Cochrane Pregnancy and Childbirth Group. No studies were retrieved from the search of great literature during the original search, therefore no updates of grey literature searches were performed. A combination of index and free-text terms were used to search, relating to the key concepts of pregnancy and antenatal care, and interventions for changing lifestyle behaviours (S2 Table). The search was limited to English language reviews published over the previous 10 years to yield primary research conducted 30+ years prior to the reviews [38]. Conference abstracts were excluded. To fulfil the requirements of an umbrella review, a systematic review search filter from the Scottish Intercollegiate Guidelines Network (SIGN) was applied. SIGN search filters are validated search filters which limit the search to a particular study type, in this case, systematic reviews.

The results were downloaded into EndNote reference management software, where they were de-duplicated. Titles and abstracts were screened using pre-defined inclusion and exclusion criteria. The full texts were retrieved and screened for all systematic reviews that had the potential to meet the inclusion criteria, or where it was not possible to definitively exclude

based on title and abstract alone. Full text screening used a pre-defined screening template developed for this review (S3 Table) and reasons for exclusion were recorded. All stages of screening were carried out by two reviewers independently (including NH, JN, LM, LA, CM, JO, LH, DJ) with any disagreements discussed and a third reviewer available for arbitration if required.

## Inclusion criteria

The inclusion criteria are based on PICOS (population, intervention, comparator, outcome, and study design). The population (P) were pregnant women at any gestational age. We did not apply restrictions based on socio-demographic factors (e.g. age, ethnicity, parity, socioeconomic status). We included reviews regardless of whether the interventions targeted specific groups of women or whether they were delivered to the general maternity population; for example, we included diet and physical activity reviews that only included interventions delivered to women with obesity, as well as those which included women with any BMI. We excluded systematic reviews reporting effectiveness of interventions delivered during the preconception or postnatal periods.

The interventions (I) included in the reviews needed to have content which aimed to change the target behaviours:

1. smoking (e.g. quit rates, relapse)

2. alcohol (e.g. level of alcohol consumption and abstinence)

3. diet (e.g. nutritional composition of the diet)

4. physical activity (e.g. levels of moderate to vigorous physical activity (MVPA))

While effective pharmacological and dietary supplement interventions (e.g. nicotine replacement therapy, folic acid supplementation) entail behavioural components for compliance, they do not primarily address the target behaviours and were thus excluded. Systematic reviews were not excluded based on the type of comparator groups (C).

The outcomes (O) of interest were the effectiveness of interventions at changing maternal target behaviours during pregnancy (i.e. smoking, alcohol consumption, diet and physical activity). No restrictions were placed on how the target behaviours were measured. Systematic reviews were excluded if they only reported the effectiveness of interventions delivered on health-related outcomes (e.g. gestational diabetes, gestational weight gain, hypertension) without reporting the effectiveness of the interventions in changing the target behaviour.

The study design (S) was limited to only include systematic reviews reporting on quantitative intervention effectiveness data, with or without meta-analysis. Primary research studies, and reviews where the primary sources of evidence were theoretical studies, qualitative data, or opinion were excluded. Mixed methods reviews were eligible for inclusion if they included quantitative effectiveness data plus another type of data; however, only the quantitative data were extracted and synthesised.

## Data extraction and quality assessment

All systematic reviews which met the inclusion criteria were data extracted using a standardised template which included: authors, year of publication, aim of the review, search strategy, inclusion and exclusion criteria, author reported conflict of interest, details of included studies (including number of included studies, publication date range, study design, population sample size, intervention location, and full citations for assessment of overlap of included studies

between systematic reviews), and narrative or meta-analysis results relating to effectiveness of interventions delivered during pregnancy on maternal behaviour outcomes.

The standardised JBI critical appraisal instrument for umbrella reviews [38] was used to assess methodological quality of included reviews (S4 Table). The checklist comprises 11 questions relating to methodological rigor, transparency of reporting and appropriateness of conclusions and recommendations, with options of "yes" if the review clearly meets the checklist criteria, and "no", "unclear" or "not applicable" if the review does not clearly meet the criteria. The reviews were awarded a score of 1 for each checklist criteria clearly met and 0 for those not met, with a maximum possible score of 11. The reviews were categorised as high quality if they scored 8–11, moderate quality for scores of 4–7, and low quality for scores of 0–3. No reviews were excluded based on quality score. The percentage of included reviews meeting the appraisal criteria was calculated for each of the 11 checklist questions.

The data extraction and quality assessment tools were piloted by a group of reviewers (NH, JN, LH, JO, LM, LA, DJ) and refined to improve consistency between reviewers. All data extraction and quality assessments were carried out by one reviewer and validated by a second reviewer (including LH, CM, LM, LA, DJ, NH, JO).

## Evidence synthesis

The evidence synthesis prioritises searching for consistency in reported effectiveness of interventions at changing women's behaviours across reviews, and identification of gaps in the existing evidence-base. A summary of existing research syntheses are presented in tabular format alongside a more detailed narrative synthesis of the systematic review characteristics and findings [38]. Summary tables include the characteristics of the included systematic reviews, quality of the reviews, overlap of included studies within reviews, and author reported conflicts of interest. The analysis of results used a systematic narrative synthesis approach [41]. Data were tabulated according to the target behaviours (i.e. smoking, alcohol, diet and physical activity) and behavioural outcomes reported (e.g. smoking abstinence, alcohol consumption, energy intake, MVPA). Reviews that conducted a meta-analysis were summarised in the tables with statistical significance and summary of the direction of effect (based on the pooled mean difference, risk ratio and odds ratio). Reviews which did not report meta-analysis and instead reported the results of studies narratively were coded to reflect the effectiveness of the interventions they reported. The code "no difference" was used if 0–33% of individual studies reported significant intervention effect (positive or negative); "inconsistent evidence" was used if 34–59% of studies reported positive or negative significant effect; and "favours intervention" or "favours control" were used if 60–100% of studies reported a positive or negative significant effect of the intervention [41, 42]. A 'Summary of Evidence' table provides a visual indication of the findings of the review using traffic light indicators [38] to indicate whether the evidence favours the intervention, control, no difference or inconsistent evidence.

# Results

## Included systematic reviews

A total of 24,388 unique records were identified by searches, of which 109 were systematic reviews of behaviour change interventions delivered in pregnancy; 16 smoking, 4 alcohol and 89 diet and/or physical activity (Fig 1). Of these, 36 reported the effectiveness of interventions delivered during pregnancy on behavioural outcomes, which are the focus of this review (S5 Table).

From these, 16 reviews reported diet and/or physical activity behaviours (18% of all diet and/or physical activity intervention reviews identified in the search).This included six reviews

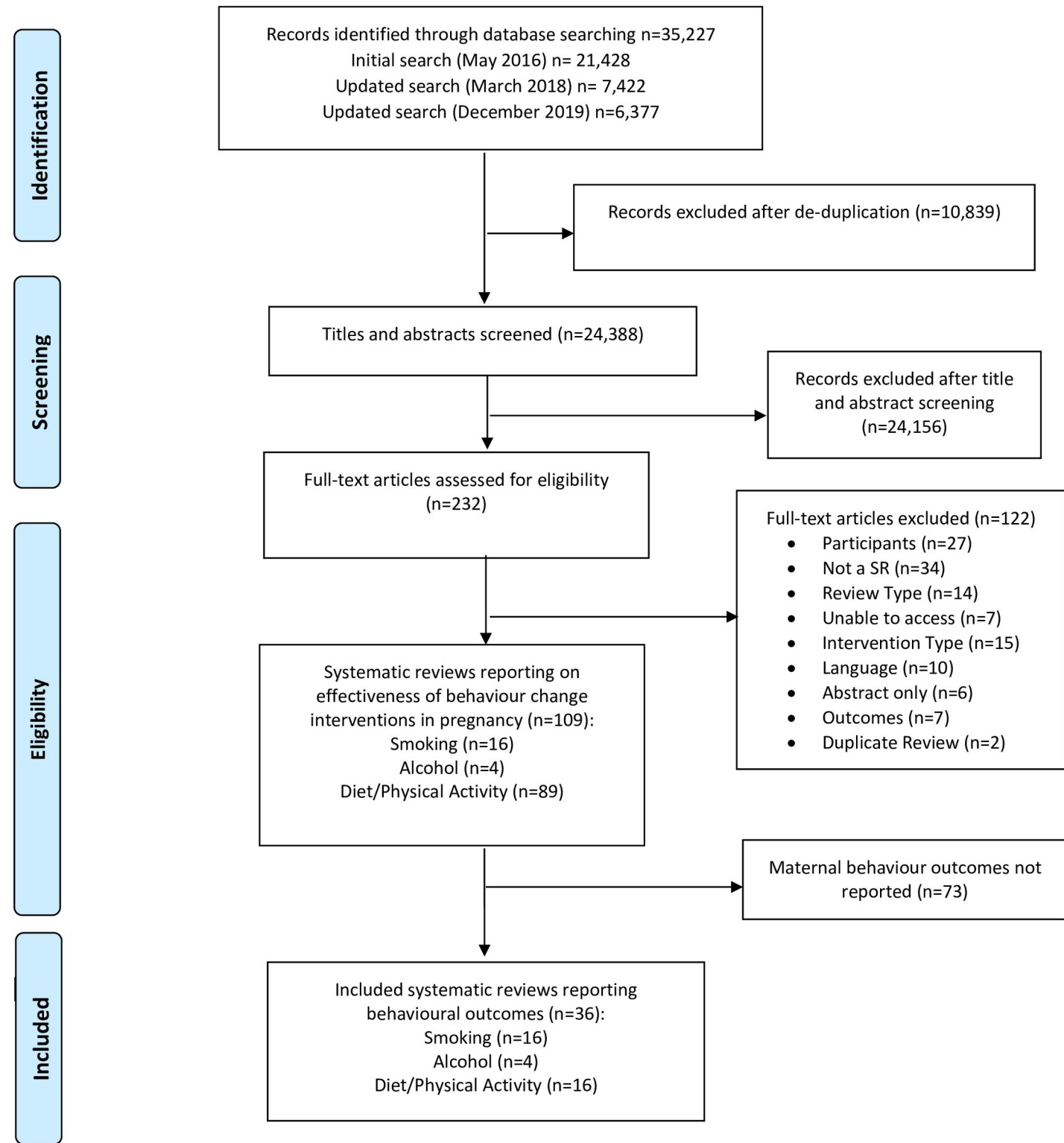

**Fig 1. PRISMA flowchart.**

reporting diet outcomes only, two reviews reporting physical activity outcomes only, and six reviews reporting a combination of physical activity and diet outcomes.

Sixteen of the included reviews were for smoking behaviours (100% of all smoking intervention reviews) and 4 for alcohol behaviours (100% of all alcohol intervention reviews) and were included in this review.

The included systematic reviews reported a range of search strategies which included a combination of searching databases alone (n = 15) and supplementing database searches with additional search strategies (n = 21); supplementary searches were predominantly carried out by diet and/or physical activity reviews (n = 15) versus other behaviours, and involved searching reference lists of included studies and/or related reviews, plus hand searching journals and clinical trials registers (Table 1, S5 Table).

The pooled sample sizes in the included reviews ranged from 0 women (empty review of alcohol interventions [44]) to 31,958 women (smoking review [45]) (Table 1, S5 Table); there was a median pooled sample size of 4,721 women in smoking reviews, 3,723 women in diet and/or physical activity reviews, and 2,047 women in alcohol reviews. Systematic reviews were restricted to RCT design in 18 reviews and included RCTs plus other study designs in 14 reviews (not reported for three smoking reviews plus one empty alcohol review). The number of intervention studies included in the systematic reviews ranged from three [46] to 88 [15], with a median of 14 studies for diet and/or physical activity and smoking, and six studies for alcohol reviews. The included studies in the reviews were published between 1952 [47] and 2018 [48].

The countries of the intervention studies included within the systematic reviews were not reported for nine of the 36 reviews (five for smoking reviews [45, 49–52], four for diet/physical activity systematic reviews [47, 53–55]) plus one empty alcohol review with no included studies to report [44]. In the remaining systematic reviews, according to the World Bank classification [43], studies were predominantly set in High-Income Countries and Upper Middle-Income Countries. Overall, 21 High-Income Countries were represented (in the 26 reviews reporting the countries) and nine Upper-Middle-Income Countries. However, Lower-Middle-Income Countries were only represented in diet/physical activity reviews (n = 1 country) and none of the reviews included any intervention studies set-in Low-Income Countries (Table 1, S5 Table).

## Quality

The quality scores of the included systematic reviews were predominantly high (67%) with no systematic reviews rated as low quality (Table 2, S6 Table). Out of a maximum possible score of 11, scores ranged from four (moderate quality) [56] to 11 (high quality) [48, 49, 57–62]. Two of the 16 smoking [48, 49], one of the four alcohol [57] and five of the 16 diet/physical activity reviews reached the maximum score of 11 [58–62]. All systematic reviews across all behaviours had clearly and explicitly stated questions (100% for question 1) and almost all scored highly for appropriate specific directives for new research (97% question 11) (Table 2). There were high scores (75–89%) for the remaining questions, with the exception of two: critical appraisal was only carried out by two researchers in 53% of systematic reviews (question 6) and the likelihood of publication bias was only assessed in 42% of systematic reviews (question 9).

## Overlap of included studies

A systematic review of systematic reviews will include duplication of original studies reported by multiple reviews. There were a total of 504 citations of included intervention studies in the

**Table 1. Summary data for included systematic reviews.**

| | Alcohol reviews (n = 4 incl. 1 empty review) | Smoking reviews (n = 16) | Diet and / or Physical Activity Reviews (n = 16) | Total across all reviews (n = 36) |
|---|---|---|---|---|
| Time period of included intervention studies in the reviews | | | | |
| Range in years: | 1982 to 2011 | 1976 to 2017 | 1952 to 2018 | 1952 to 2018 |
| Earliest year before 1990 (n) | 1 | 2 | 2 | 5 |
| Earliest year from 1990 (n) | 1 | 8 | 1 | 10 |
| Earliest year from 2000 (n) | 1 | 2 | 11 | 14 |
| Earliest year from 2010 (n) | 0 | 4 | 2 | 6 |
| Search strategies: | | | | |
| Databases only (n) | 2 | 12 | 1 | 15 |
| Databases + Supplementary searches (n) | 2 | 4 | 15 | 21 |
| Included study designs: | | | | |
| RCT only (n) | 1 | 6 | 11 | 18 |
| RCT + other design (n) | 2 | 7 | 5 | 14 |
| Not reported (n) | 0 | 3 | 0 | 3 |
| Number included studies in the systematic reviews | | | | |
| Range (n) | 0 to 10 | 3 to 88 | 3 to 65 | 3 to 88 |
| Median (n, IQR) | 6 (4 to 8) | 14 (8 to 25) | 14 (8 to 22) | |
| Not reported (n) | n/a | n/a | n/a | n/a |
| Pooled sample sizes included in the systematic reviews | | | | |
| Range (n women) | 0 to 3,494 | 1,009 to 31,958 | 209 to 15,328 | 0 to 31,958 |
| Median (n women, IQR) | 2,047 (536 to 3,408) | 4,721 (1,009 to 15,338) | 3,723 (1,457 to 5,693) | |
| Not reported (n) | n/a | 1 review | 1 review | 2 reviews |
| Countries of intervention studies included in the systematic reviews (reported for n systematic reviews) | Reported for 3 out of 4 SRs (4th empty review) | Not reported for 5 SRs | Not reported for 4 SRs | Not reported for 10 SRs (incl. 1 empty SR) |
| HICs represented in the included interventions | 1. Norway | 1. Argentina | 1. Australia | 1. Argentina |
| | 2. Sweden | 2. Australia | 2. Belgium | 2. Australia |
| | 3. UK | 3. Canada | 3. Canada | 3. Belgium |
| | 4. USA | 4. Denmark | 4. Croatia | 4. Canada |
| | | 5. France | 5. Denmark | 5. Croatia |
| | | 6. Greece | 6. Finland | 6. Denmark |
| | | 7. Netherlands | 7. Germany | 7. Finland |
| | | 8. New Zealand | 8. Ireland | 8. France |
| | | 9. Norway | 9. Italy | 9. Germany |
| | | 10. Poland | 10. Netherlands | 10. Greece |
| | | 11. Spain | 11. Norway | 11. Ireland |
| | | 12. Sweden | 12. New Zealand | 12. Italy |
| | | 13. UK | 13. Spain | 13. Netherlands |
| | | 14. USA | 14. Sweden | 14. New Zealand |
| | | | 15. Taiwan | 15. Norway |
| | | | 16. UK | 16. Poland |
| | | | 17. USA | 17. Spain |
| | | | | 18. Sweden |
| | | | | 19. Taiwan |
| | | | | 20. UK |
| | | | | 21. USA |

*(Continued)*

**Table 1.** (Continued)

| | Alcohol reviews (n = 4 incl. 1 empty review) | Smoking reviews (n = 16) | Diet and / or Physical Activity Reviews (n = 16) | Total across all reviews (n = 36) |
|---|---|---|---|---|
| UMICs represented in the included interventions | 1. Mexico | 1. Brazil | 1. Brazil | 1. Brazil |
| | | 2. Cuba | 2. China | 2. China |
| | | 3. Mexico | 3. Columbia | 3. Columbia |
| | | | 4. Iran | 4. Cuba |
| | | | 5. South Africa | 5. Iran |
| | | | 6. Thailand | 6. Mexico |
| | | | 7. Turkey | 7. South Africa |
| | | | | 8. Thailand |
| | | | | 9. Turkey |
| LMICs represented in the included interventions | None | None | 1. Egypt | 1. Egypt |
| LICs represented in the included interventions | None | None | None | None |

Income status of the countries defined according to the World Bank data for the current 2020 fiscal year "low-income economies are defined as those with a GNI per capita, calculated using the World Bank Atlas method, of $1,025 or less in 2018; lower middle-income economies are those with a GNI per capita between $1,026 and $3,995; upper middle-income economies are those with a GNI per capita between $3,996 and $12,375; high-income economies are those with a GNI per capita of $12,376 or more" [43]

Abbreviations: HICs = High Income Countries; UMICs = Upper Middle-Income Countries; LMICs - = Lower Middle-Income Countries; LICs = Lower Income Countries.

16 smoking reviews; after removal of duplicate citations of the same publications across multiple reviews there were 298 unique publications remaining (S7 Table). The 16 diet and/or physical activity reviews similarly had a total of 491 citations, of which 311 were unique publications. The four alcohol reviews had 26 citations for included interventions, of which 16 were unique publications.

**Smoking behaviours.** Sixteen systematic reviews reported behaviour outcomes for smoking interventions during pregnancy [15, 45, 48–52, 56, 63–70]. The outcomes reported were smoking abstinence (or cessation), smoking relapse, and smoking reduction. These were

**Table 2. Summary of quality assessment of included reviews.**

| Behaviour group | Quality assessment question | | | | | | | | | | | Total score | Quality category |
|---|---|---|---|---|---|---|---|---|---|---|---|---|---|
| | 1 | 2 | 3 | 4 | 5 | 6 | 7 | 8 | 9 | 10 | 11 | | |
| **Subtotal alcohol reviews** | 100% (4/4) | 100% (4/4) | 100% (4/4) | 75% (3/4) | 75% (3/4) | 75% (3/4) | 75% (3/4) | 75% (3/4) | 25% (1/4) | 50% (2/4) | 100% (4/4) | Range 7–11 | 50% moderate 50% high |
| **Subtotal smoking reviews** | 100% (16/16) | 81% (13/16) | 81% (13/16) | 81% (13/16) | 75% (12/16) | 56% (9/16) | 81% (13/16) | 88% (14/16) | 38% (6/16) | 75% (12/16) | 94% (15/16) | Range 4–11 | 68.7% high 31.3% medium |
| **Subtotal diet and/or physical activity reviews** | 100% (16/16) | 94% (15/16) | 88% (14/16) | 88% (14/16) | 75% (12/16) | 44% (7/16) | 75% (12/16) | 75% (12/16) | 50% (8/16) | 94% (15/16) | 100% (16/16) | Range 6–11 | 68.7% high 31.3% medium |
| **Total all behaviours** | **100% (36/36)** | **89% (32/36)** | **86% (31/36)** | **83% (30/36)** | **75% (27/36)** | **53% (19/36)** | **81% (29/36)** | **83% (30/36)** | **42% (15/36)** | **81% (29/36)** | **97% (35/36)** | Range 4–11 | 67% high 33% moderate |

quality assessment questions were: 1) Is the review question clearly and explicitly stated?; 2) Were the inclusion criteria appropriate for the review question?; 3) Was the search strategy appropriate?; 4) Were the sources and resources used to search for studies adequate?; 5) Were the criteria for appraising studies appropriate?; 6) Was critical appraisal conducted by two or more reviewers independently?; 7) Were there methods to minimize errors in data extraction?; 8) Were the methods used to combine studies appropriate?; 9) Was the likelihood of publication bias assessed?; 10) Were recommendations for policy and/or practice supported by the reported data?; 11) Were the specific directives for new research appropriate?. For total score 1 is given if yes otherwise it is zero. For quality: Low quality is 0–3. Moderate quality is 4–7. High quality is 8–11.

measured by self-reporting, mean number of cigarettes per day, biochemically validated reduction, or mean biochemical cotinine (S8 Table, S9 Table).

**Smoking abstinence (or cessation) in pregnancy.** Smoking abstinence (or cessation) during pregnancy was reported in 14 systematic reviews [15, 45, 48–52, 56, 63–66, 69, 70] (S8 Table, S9 Table and Table 3), including nine reviews reporting 33 meta-analyses [15, 48, 49, 52, 63–65, 69, 70] (S8 Table and Table 3). Findings from meta-analyses were inconsistent. Sixteen meta-analyses identified a significant increase in smoking abstinence during pregnancy in the intervention arms with RRs ranging from 1.23 (95% CI 1.04, 1.45) [15] to 4.39 (95% CI 1.89, 10.21) [15] (S8 Table). However, 17 meta-analyses reported no significant difference between intervention and control arms (S8 Table). There was a noticeable difference in the number of studies included in meta-analyses reporting significant and non-significant results; all non-significant meta-analyses included data from seven or fewer studies, whereas the significant results were predominantly from meta-analyses including greater than 10 studies (n = 12 out of 16). The largest meta-analysis (n = 97 studies) compared interventions for smoking cessation with control groups and reported a significant increase in smoking cessation in late pregnancy (RR 1.35, 95% CI 1.23, 1,48) [15] (S8 Table and Table 3). Results from narrative reviews were not consistent, with some favouring the intervention [45, 51, 63], while others showing no difference or inconsistent evidence between interventions [51, 56, 67].

**Smoking abstinence (or cessation) postpartum.** The effect of smoking interventions delivered in pregnancy on smoking abstinence postpartum was explored by four reviews reporting 20 meta-analyses [15, 52, 64, 65]. Findings largely represented a lack of statistically significant smoking abstinence in the postpartum period following interventions being delivered during (17 out of 20 meta-analyses, S8 Table). One narrative synthesis reported higher postpartum abstinence in the intervention group [56] (Table 3, S9 Table).

**Smoking relapse.** Smoking relapse during pregnancy was measured in one meta-analysis and one narrative review [63, 64], and in one narrative synthesis [51] with no significant changes in smoking relapse in pregnancy reported (S8 and S9 Tables).

**Smoking reduction.** Two systematic reviews reported 13 meta-analyses of smoking reduction, or number of cigarettes smoked per day, during pregnancy and postnatally [15, 64]. There were mixed results. Seven meta-analyses reported a significant improvement ranging from RR 1.35 (95% CI 1.07, 1.71) to 1.66 (95% CI 1.27, 2.17) for reduction in smoking in women in the intervention arms, and a SMD of -0.25 (95% CI -0.46, -0.03) to -0.55 (95% CI -0.94, -0.15) cigarettes per day (S8 Table). However, six meta-analyses reported no significant difference (S8 Table). Results from systematic reviews which reported results of biochemical tests to measure smoking behaviour showed a lack of statistical significance in five out of six meta-analyses reported by two systematic reviews [15, 64] (S8 Table).

## Alcohol behaviours

Three systematic reviews reported data on alcohol behaviours, including alcohol consumption and abstinence [57, 71, 72]. All systematic reviews reported narrative synthesis of interventions (Table 4, S10 Table).

**Alcohol consumption.** All reviews reported data for alcohol consumption; two found no difference in the quantity of alcohol consumed between intervention and control arms of included studies [57, 72], whereas one systematic review reported results that favoured the intervention [71] (S10 Table).

**Alcohol abstinence.** Two systematic reviews reported data for abstinence of alcohol during pregnancy [57, 72]; both reviews reported inconsistent evidence between the included intervention studies (S10 Table).

Table 3. Evidence summary table for systematic reviews reporting maternal smoking behaviour.

| Review | Smoking abstinence (or cessation) pregnancy | Smoking abstinence (or cessation) postpartum | Smoking relapse pregnancy | Smoking relapse postpartum | Smoking reduction (self-report) | Smoking reduction (mean cigarettes/day) | Smoking reduction (biochemically validated) | Smoking reduction (mean cotinine) |
|---|---|---|---|---|---|---|---|---|
| Agboola *et al.* 2010 [63] | ↑ 6–9 mos (n = 2)<br>↑ end preg (n = 5)<br>↔ (n = 1) | | ↔ med term (n = 3)<br>↔ long term (n = 1) | | | | | |
| Chamberlain *et al.* 2013 [64] | ↑ late preg (n = 27)[a]<br>↑ late preg bioch (n = 18)[a]<br>↑ late preg (n = 16)[b]<br>↑ late preg bioch (n = 12)[b] | ↑ >18 mos (n = 2)[a]<br>↑ 6–11 mos (n = 3)[b] | ↑ late preg (n = 8)[a] | | ↑ late preg (n = 2)[a] | ↓ late preg (n = 9)[a] | ↑ late preg (n = 3)[a] | ↓ late preg (n = 3)[a] |
| Chamberlain *et al.* 2013 [64] | ↑ late preg spon quit (n = 4)[b] | ↑ 12–17 mos (n = 2)[b] | | | | | | |
| Chamberlain *et al.* 2017 [15] | ↑ late preg (n = 30)[c]<br>↑ late preg bioch (n = 21)[c]<br>↑ late preg (n = 18)[d]<br>↑ late preg bioch (n = 15)[d]<br>↑ late preg (n = 2)[h]<br>↑ late preg (n = 4)[i]<br>↑ late preg bioch (n = 97)[k] | ↑ 0–5 mos (n = 11)[c]<br>↑ 0–5 mos (n = 2)[e] | | | ↑ late preg (n = 5)[c]<br>↑ late preg (n = 2)[d] | ↓ late preg (n = 2)[c]<br>↓ late preg (n = 2)[i] | ↓ late preg (n = 2)[c] | ↓ late preg (n = 6)[c] |
| Chamberlain *et al.* 2017 [15] | ↑ late preg spon quit (n = 5)[d]<br>↑ late preg bioch (n = 3)[e]<br>↑ late preg (n = 4)[f]<br>↑ late preg bioch (n = 3)[f]<br>↑ late preg (n = 3)[g]<br>↑ late preg bioch (n = 3)[g]<br>↑ late preg (n = 4)[j]<br>↑ late preg (n = 7)[h]<br>↑ late preg bioch (n = 6)[h]<br>↓ late preg (n = 2)[i]<br>↑ late preg (n = 3)[j]<br>↑ late preg bioch (n = 2)[j] | ↑ 6–11 mos (n = 4)[c]<br>↑ 12–17 mos (n = 3)[c]<br>↓ >18 mos (n = 3)[c]<br>↑ 0–5 mos (n = 6)[d]<br>↑ 6–11 mos (n = 4)[d]<br>↑ 0–5 mos (n = 2)[h]<br>↑ 0–5 mos (n = 3)[i]<br>↓ 6–11 mos (n = 3)[i]<br>↑ 0–5 mos (n = 2)[h]<br>↑ 6–11 mos (n = 3)[h]<br>↓ 0–5 mos (n = 2)[j] | | | | ↓ late preg (n = 2)[d] | ↑ late preg (n = 2)[d] | ↓ late preg (n = 2)[h] |
| Filion *et al.* 2011 [65] | ↑ 27–37 wk (n = 7) | ↑ 28 wk gest to 6 wk post (n = 6) | | | | | | |
| Griffiths *et al.* 2018 [48] | ↑ (n = 12) | | | | | | | |
| Hand *et al.* 2017 [50] | ↑ (n = 14) | | | | | | | |
| Hemsing *et al.* 2012 [67] | ↓↑ (n = 4) | | | | | | | |

*(Continued)*

**Table 3.** (Continued)

| Review | Smoking abstinence (or cessation) pregnancy | Smoking abstinence (or cessation) postpartum | Smoking relapse pregnancy | Smoking relapse postpartum | Smoking reduction (self-report) | Smoking reduction (mean cigarettes/day) | Smoking reduction (biochemically validated) | Smoking reduction (mean cotinine) |
|---|---|---|---|---|---|---|---|---|
| **Heminger et al. 2016 [66]** | ↔ (n = 7) | | | | | | | |
| **Hettema et al. 2010 [49]** | ↑$^{<6\text{ mos}}$ (n = 7)<br>↑$^{>6\text{ mos}}$ (n = 2) | | | | | | | |
| **Hubbard et al. 2016 [68]** | ↔ (n = 3) | | | | | | | |
| **Kintz et al. 2014 [45]** | ↑ (n = 24) | | | | | | | |
| **Naughton et al. 2008 [69]** | ↑ (n = 12)[l]<br>↑ (n = 7)[m] | | | | | | | |
| **Su et al. 2014 [51]** | ↑ (n = 3)[n] | | | ↔ $^{\text{long term}}$ (n = 32) | | | | |
| **Veisani et al. 2017 [70]** | ↑ (n = 3) | | | | | | | |
| **Washio et al. 2016 [56]** | ↓↑ (n = 9) | ↑$^{6\text{wk-6mos}}$ (n = 4) | | | | ↑ (n = 1) | | ↔ (n = 2) |
| **Wilson et al. 2018 [52]** | ↑ (n = 21) | ↑$^{\text{early post}}$ (n = 7)<br>↑$^{\text{late post}}$ (n = 8) | | | | | | |

Bold = meta-analysis data

Abbreviations: wk = weeks, mos = months, med = medium, gest = gestational, post = postpartum, bioch = biochemically validated, spon quit = spontaneous quitter.

a: counselling vs. usual care [64]

b: counselling vs. less intense interventions [64]

c: counselling vs. usual care [15]

d: counselling vs. less intense interventions [15]

e: health education vs. usual care [15]

f: health education vs. less intense intervention [15]

g: feedback vs. less intense intervention [15]

h: incentive vs. usual care [15]

i: incentive vs. alternative intervention [15]

j: incentive vs. less intense intervention [15]

k: Social support vs. less intense intervention [15]

l: Maternal health intervention vs. usual care [15]

m: Maternal health intervention vs. less intense intervention [15]

n: Smoking cessation intervention vs. control [15]

o: Self-help vs. usual care [69]

p: Self-help vs. less intensive self-help [69]

q: Incentives [51]

Key

Inconsistent evidence (for either direction of effect or statistical significance)

Inconsistent evidence (for either direction of effect or statistical significance)

Favours intervention

Favours control

Outcome not reported

↑ Direction of effect—increased

↓ Direction of effect—decreased

↓↑ Direction of effect—mixed results

↔ Direction of effect—no difference

**Table 4. Evidence summary table for systematic reviews reporting maternal alcohol behaviours.**

| Review | Alcohol consumption | Alcohol abstinence |
|---|---|---|
| Gilinsky *et al.* 2011 [72] | ↔ (n = 8) | ↔ ↑ (n = 6) |
| Gebara *et al.* 2013 [71] | ↓ (n = 8) | |
| Stade *et al.* 2009 [57] | ↔ (n = 4) | ↔ ↑ (n = 4) |

Key

Inconsistent evidence (for either direction of effect or statistical significance)

No significant difference between intervention and control arm

Favours intervention

Outcome not reported

↑ Direction of effect—increased

↓ Direction of effect—decreased

↔ Direction of effect—no difference

## Dietary behaviours

Twelve reviews reported data on dietary behaviours including energy intake, macronutrients, micronutrients, and "healthy" dietary behaviours or patterns (Table 5). Four reviews primarily reported meta-analysis (although for some outcomes the data reported were the results of single intervention studies) [59, 60, 62, 73] (S11 Table) and eight reviews reported narrative synthesis [46, 47, 53, 58, 61, 74–76] (S12 Table).

**Energy intake.** Six systematic reviews reported data for energy intake [47, 58–60, 73, 74]. Three systematic reviews with meta-analysis results reported energy intake outcomes [59, 60, 73]; however, the majority of outcomes were reported from single study data (S11 Table). Muktabhant *et al.* [60] reported meta-analysis of 14 studies which showed decreased energy intake (MD -570.77kj/day). Lau *et al.* [59] reported the difference in kcal/day measured by both self-report and *"software methods"*, at different gestations (during pregnancy at 15 to 18 weeks and 27 to 28 weeks, and postnatally at 12–20 weeks and 12 months). The majority of studies showed decreased energy intake in the intervention group; however, only software measured energy intake at 27 to 28 weeks in pregnancy, and both postnatal time points were significantly reduced (MD -167.00, -305.28, and -495.00 k/cal per day respectively) [59]. Sherifali *et al.* [73] reported single study data for decreased total energy intake, and increased percent energy from carbohydrate and protein sources in the intervention arm, although the results were not statistically significant. Three systematic reviews reporting narrative synthesis showed mixed results (S12 Table): one review included five studies that favoured the intervention overall [58], one included two studies that reported inconsistent evidence [74] and one reported no difference between intervention and control arms among five studies [47].

**Macronutrients.** Four reviews reported data for fat intake; one review reported multiple results from single studies [59] (S11 Table) and three reported a narrative synthesis [58, 74, 76] (S12 Table). Lau *et al.* [59] reported four results on the effectiveness of interventions at reducing saturated fat intake, either self-reported or measured using software, at 15 to 18 and 27 to 28 weeks gestation. All showed decreased saturated fat intake; however, only software measure of fat intake at 27 to 28 weeks was significantly reduced by -4.40 g per day [95% CI -5.63, -3.17]. The narrative syntheses all favoured the intervention for the reduction of fat, saturated fat and solid fat intake.

Three systematic reviews reported narrative synthesis of protein intake [47, 58, 74] (S12 Table). Two reviews favoured the intervention and reported increased protein intake compared with controls [47, 74], whereas one review found no difference between groups [58].

**Table 5. Evidence summary table for systematic reviews reporting maternal diet behaviours.**

| Review | Energy | Fat | Carbohydrate | Low GI diet or GI/GL | Protein | Fibre | Micronutrient | Diet Behaviours | Fruit and vegetables | Animal-based food |
|---|---|---|---|---|---|---|---|---|---|---|
| **Bain et al. 2015 [58]** | ↓(n = 5) | ↓ (n = 5) | ↓(n = 5) | | ↔ (n = 3) | ↔↑ (n = 2) | ↔ (n = 1) | ↑H (n = 3) | | |
| **Brown et al. 2012 [53]** | | | | | | | | ↑H (n = 1) | | |
| **Flynn et al. 2016 [74]** | ↓↑ (n = 2) | ↓ (n = 5) | ↓(n = 2) | ↓↑GI/GL (n = 2) | ↑ (n = 2) | ↔ ↑ (n = 2) | ↑ (n = 1) | ↑H (n = 3) ↑S (n = 1) | ↑ (n = 4) | ↔ (n = 1) |
| **Gardner et al. 2011 [75]** | | | | | | | | ↔H (n = 6) | | |
| **Webb-Girard et al. 2012 [47]** | ↔ (n = 5) | | | | ↑ (n = 7) | | ↑ (n = 4) | ↑H (n = 4) ↑S (n = 1) | ↑ (n = 1) | ↑ (n = 1) |
| **Mohd Yusof et al. 2014 [46]** | | | | ↓ GI/GL (n = 3) | | | | | | |
| **Muktabhant et al. 2015 [60]** | ↓(n = 14) | | | | | ↑(n = 8) | | | | |
| **O'Brien et al. 2014 [76]** | | ↓ (n = 1) | ↓(n = 1) | | | ↓R (n = 1) ↑W (n = 1) | | ↓U FR (n = 1) ↓U FA (n = 1) | ↑FV (n = 1) ↑AN (n = 1) | ↓M (n = 1) ↑F (n = 1) |
| **Lau et al. 2017 [59]** | ↓ (n = 1 or 2 studies reporting 8 energy outcomes; 3Sig, 5NS) | ↓(n = 1 study reporting 4 fat outcomes, 1Sig, 3NS) | | | | | | | ↑(n = 14) | |
| **Shepherd et al. 2017 [61]** | | | | | | | | ↑H (n = 17) | | |
| **Sherifali et al. 2017 [73]** | ↓T (n = 1) ↑C (n = 1) ↑P (n = 1) | | | | | | | | | |
| **Tieu et al. 2017 [62]** | | | | ↑D (n = 1) | | | | ↓H (n = 1) ↑H (n = 1) | | |

Bold = meta-analysis data; Abbreviations: AN = avocado and nuts, C = energy from carbohydrate, D = Low GI diet, F = fish, FA = fast food, FR = fried food, FV = fruit and vegetables, GI/GL = glycaemic index or glycaemic load, H = healthy, M = meat, NS = not significant, P = energy from protein, R = refined grain, S = snacks, Sig = significant, T = total energy, U = unhealthy, W = wholegrain

Key

Inconsistent evidence (for either direction of effect or statistical significance)

No significant difference between intervention and control arm

Favours intervention

Favours control

Outcome not reported

↑ Direction of effect—increased

↓ Direction of effect—decreased

↓↑ Direction of effect—mixed results

↔ Direction of effect—no difference

The evidence base relating to fibre was inconsistent. One meta-analysis of eight studies reported an increased consumption of fibre [60] (S11 Table), although this was not statically significant (RR 1.53, 95% CI 0.94, 2.1). Two reviews reporting narrative synthesis showed inconsistent evidence relating to overall fibre intake [58, 74], whereas one review reported a

significant increase in wholegrain and decrease in refined grain consumption [76] (S12 Table); however, the significant results were from single studies.

Three reviews reported narrative synthesis on carbohydrate/sugar intake [58, 74, 76] (S12 Table). Two reviews favoured the intervention in reducing sugary foods [58, 76], whereas two reported inconsistent evidence or no difference in relation to overall carbohydrate [58, 74]. Three reviews reported glycaemic index/load or following a low GI diet [46, 62, 74]. A single study in a meta-analysis reported significantly increased consumption of a low GI diet following intervention (RR 5.37, 95% CI 1.93, 14.89; S11 Table). One narrative synthesis review identified inconsistent evidence [74] while the other favoured the intervention [46] (S12 Table).

**Micronutrients.**   Three reviews reported narrative synthesis data for dietary sources of micronutrients [47, 58, 74], although two of these reported data from single studies only [58, 74] (S12 Table). Bain *et al.* [58] reported no difference in iron intake in a single study, whereas Webb-Girard *et al.* [47] reported data from four studies in low income countries that, overall, favoured the intervention for increased iron intake. Single study data also reported a significant increase in calcium [74], but no difference in vitamin D or folate [58].

**Dietary behaviours/patterns.**   One review [62] reported single study results for following weight reducing diets or making dietary changes since the start of intervention; neither were statistically significant (S11 Table). Six reviews reported narrative synthesis of dietary behaviours measured in a variety of ways [47, 53, 58, 61, 74, 75] (S12 Table). Five reviews favoured the intervention arm overall in improving healthy dietary behaviours [47, 53, 58, 61, 74], whereas one review reported no difference in dietary behaviours [75].

All reviews reporting fruit and vegetable intake reported data that favoured the intervention arm. A meta-analysis of 14 studies identified a significantly increased consumption of fruit and vegetable servings per day at 30–36 weeks gestation among women receiving the intervention (MD 0.27, 95% CI 0.01, 0.53) [59] (S11 Table). Three reviews reporting narrative synthesis also all favoured the intervention for both fruit and vegetable consumption [47, 74, 76] (S12 Table).

Three reviews reported narrative synthesis for animal-based food sources including meat and/or fish [47, 74, 76], although the syntheses were either results from a single study or from two studies (S12 Table). One review identified no difference between intervention and control arms [74], whereas two reviews favoured the intervention for reducing meat and increasing fish consumption in HICs [76], or increasing meat consumption in LMICs [47]. It is important to note that the aim of interventions to either reduce or increase animal-based food sources is likely to differ based on geography and country income status.

Two reviews reported significant increases in snack consumption following intervention from single studies [47, 74], and one review reported significantly decreased fried food and fast food consumption from single studies [76] (S12 Table).

## Physical activity behaviours

Ten reviews reported physical activity behaviours [54, 55, 58, 59, 61, 62, 73, 75, 77, 78]. The behaviours reported varied across meta-analyses and narrative reviews, and included moderate to vigorous physical activity (MVPA, min/week), steps (steps/day), VO$_2$ max (amount of oxygen used during exercise), attending a gym, a combination of mixed measures of physical activity (e.g. physical activity index, engagement in leisure time physical activity, metabolic equivalent (MET)) (Table 6).

Three systematic reviews reported meta-analyses of physical activity data [59, 62, 78] (S13 Table). Interventions did not appear to impact on MVPA or steps during pregnancy [59, 78], but significantly increased mean METs (SMD 0.39, 95% CI 0.14, 0.64) and VO2 max (SMD

**Table 6. Evidence summary table for systematic reviews reporting maternal physical activity.**

| Review | Moderate to vigorous physical activity (min/week) | METs (min/week) | Steps (steps/day) | VO$_2$ max | Attending gym at 3 months postnatal | Physical activity (mixed measures) |
|---|---|---|---|---|---|---|
| **Bain** *et al.* **2015** [58] | | | | | | ↓↑ (n = 8) |
| **Currie** *et al.* **2013** [54] | | | | | | ↑ (n = 10) |
| **Gardner** *et al.* **2011** [75] | | | | | | ↓↑ (n = 4) |
| **Lau** *et al.* **2017**[a] [59] | ↑ ante 24–26 wk (n = 1) / ↑ ante 27–28 wk (n = 1) / ↑ ante 34–36 wk (n = 1) / ↑ post 12 mos (n = 1) | | ↑ ante 12–28 wk (n = 3) / ↑ ante 24–30 wk (n = 3) / ↑ ante 32–36 wk (n = 2) / ↑ post 6 wk (n = 1) / ↑ post 12 mos (n = 1) | | | |
| **Lau** *et al.* **2017**[b] [59] | ↑ post 6 wk (n = 1) / ↑ post 13 wk (n = 1) / ↑ post 12 mos (n = 1) | | | | | |
| **Nascimento** *et al.* **2012** [55] | | | | | | ↑ (n = 1) |
| **Shepherd** *et al.* **2017** [61] | | | | | | ↑ (n = 7) |
| **Sherifali** *et al.* **2017** [73] | | | | | | ↔ (n = 1) |
| **Tieu** *et al.* **2017** [62] | | | | | ↓ (n = 1) | |
| **Chan** *et al.* **2019** [77] | | | | | | ↑ (n = 9) |
| **Flannery** *et al.* **2019** [78] | | ↑ (n = 8) | ↓ (n = 3) | ↑ (n = 2) | | |

Bold = meta-analysis data; Abbreviations: wk = weeks, mos = months, ante = antenatal, post = postpartum, sr = self-reported, ob = objectively measured

a and b are the same systematic review with but with different findings (e.g. significant or no significant) according to the time of measuring (i.e. pregnancy weeks)

Key

Inconsistent evidence (for either direction of effect or statistical significance)

No significant difference between intervention and control arm

Favours intervention

Favours control

Outcome not reported

↑ Direction of effect—increased

↓ Direction of effect—decreased

↓↑ Direction of effect—mixed results

↔ Direction of effect—no difference

0.55, 95% CI 0.34, 0.75) [78]. Seven systematic reviews reported a narrative synthesis (Table 6, S14 Table) and used a range of physical activity measurements [54, 55, 58, 61, 73, 75, 77]. Five narrative syntheses favoured the intervention and reported increased levels of physical activity, or a reduced decline in physical activity levels over the course of pregnancy, among women in intervention arms, while two reported no difference and one reported inconsistent evidence.

Interventions delivered during pregnancy revealed mixed results for level of postpartum physical activity, depending on the method of measurement. MVPA (reported by single studies) significantly increased at 6 and 13 weeks, and 12 months, postnatal when physical activity

was assessed using self-reported measurements (S13 Table). However, no significant difference was observed for MVPA or steps when physical activity was measured objectively (i.e. accelerometer or pedometer) (Table 6, S13 Table).

## Conflict of interest

Given the nature of the topics of the reviews and potential conflicts of interest, particularly relating to industry funding, we have summarised whether any conflicts of interest were reported by review authors (S15 Table). Out of the 36 included studies, 30 included a conflict of interest statement. All alcohol and diet/physical activity systematic reviews included a statement regarding conflict of interest, whereas there were only 10 (out of 16) smoking reviews that included a conflict of interest statement. Of the 30 reviews that made a declaration of potential conflicts of interest, 24 reported that there were no conflicts of interest (all alcohol reviews, seven smoking reviews, 13 diet and/or physical activity reviews). Potential conflicts of interests declared were related to receiving funding (including unrelated activities to the review) from organisations with a potential conflict of interest (e.g. production of nicotine replacement therapy); having stock options in a company; working on other projects of a similar nature; and investigators on trials that were screened and included or excluded from the review (S15 Table).

## Discussion

This systematic review of systematic reviews provides an overview of the existing evidence base on the effectiveness of interventions at changing pregnant women's behaviours in the context of smoking, alcohol consumption, diet and physical activity. When considering all evidence reported, there was a consistent pattern across all reviews reporting that women in the intervention arms increased fruit and vegetable consumption and decreased their carbohydrate intake. Overall, there was also fairly consistent evidence across the majority of reviews for improving healthy diet behaviours, reducing fat intake, and reduction in smoking during pregnancy. However, there was a lack of consistent evidence across reviews reporting improvements from interventions in relation to energy, protein, fibre, or micronutrient intakes; smoking cessation, abstinence and relapse; any alcohol or physical activity behaviours.

The most robust evidence is from the reviews which reported meta-analyses and pooled data from >1 study. These meta-analyses demonstrated some similarities and some differences to the overall collective of all reviews. There was a significantly decreased energy intake [59, 60] and increased fruit and vegetable consumption [59] among women in the intervention arms in all meta-analyses reported for these outcomes. There were multiple meta-analyses of >1 study for smoking behaviours: approximately half of meta-analyses for smoking abstinence or cessation during pregnancy found significant improvements in the intervention arm (16 out of 33 analyses reported by six reviews) [15, 48, 52, 64, 69, 70] with meta-analyses of a greater number of studies more likely to report a significant improvement, and the majority of meta-analyses (eight out of 13) for smoking reduction in pregnancy showed significant improvements in the intervention arm. There was limited significant effect on postpartum smoking abstinence or cessation when interventions were delivered during pregnancy (three out of 20 meta-analyses) reported by two reviews [15, 52]. For physical activity, meta-analyses of >1 study identified a significant increase in METs and VO2 max [78], but no significant difference in steps per day [59, 78]. No alcohol studies were pooled in a meta-analysis. When considering the narrative syntheses which included >1 study there were mixed results. The reviews reporting narrative syntheses of diet behaviours demonstrated the most promising results with 15 out of 22 syntheses favouring the intervention arms [46, 47, 58, 61, 74]. Three

out of five syntheses of >1 study favoured the intervention for physical activity behaviours [54, 61, 77]. For smoking behaviours, there were 12 syntheses of >1 study of which four favoured the intervention arms [45, 50, 51, 56]. One out of five syntheses of >1 study for alcohol behaviours favoured the intervention [71].

Comparing the volume of evidence across the different behavioural domains, there were the same number of systematic reviews that focussed on smoking behaviours and diet and/or physical activity behaviours (n = 16), which had a similar number of total citations (n = 504 and 491) and unique publications which contributed to these reviews (n = 298 and 311). However, there is a clear gap in the evidence-base of systematic reviews (and interventions to inform systematic reviews) relating to alcohol consumption with only four reviews reporting 16 unique publications out of 26 total citations. Data were most comprehensive for systematic reviews of smoking interventions which primarily reported meta-analyses rather than narrative synthesis, although the majority of meta-analyses did originate from two large systematic reviews which reported extensive meta-analyses of multiple smoking behaviour outcomes according to sub-groups of type of intervention and measurement methods [15, 64]. The evidence for diet and physical activity behaviours included approximately equal data from systematic reviews reporting meta-analyses and narrative syntheses. However, in the reviews reporting meta-analyses there was frequent reporting of behavioural outcomes from single studies rather than pooled data from multiple studies which limits the evidence base. With dietary behaviours, there was an exception for fruit and vegetable consumption, energy intake and fibre where robust meta-analyses were reported. Whereas, the evidence-base presented in the narrative syntheses included multiple studies for most outcomes (e.g. carbohydrates, fats and protein). This suggests that evidence exists for these behavioural outcomes, but that they are measured and reported inconsistently and the heterogeneity limits the ability for data to be pooled in order to provide a robust overview of the evidence-base.

When comparing the volume of available systematic review evidence across behavioural domains, it is also important to consider the proportion of existing systematic reviews on behavioural interventions that report behavioural outcomes. When searching for evidence, there were a considerably greater number of systematic reviews of diet and/or physical activity interventions compared with smoking and alcohol (n = 89, 16 and four respectively). While 100% of smoking and alcohol systematic reviews reported smoking and alcohol behaviours, only 18% of diet and/or physical activity reviews reported these behaviours, and the majority focussed on the effectiveness of behaviour change intervention on health-related outcomes (e.g. gestational diabetes, large for gestational age babies). While the health-related outcomes of behavioural interventions are undoubtedly important from a clinical perspective, we should not underestimate the importance of measuring behaviours, both from a public health perspective where behaviour change is considered an outcome in its own right, and from a mechanistic perspective to understand the processes involved in interventions having an impact on health outcomes. For example, if diet and physical activity interventions are not effective at reducing the risk of adverse health outcomes but the behaviour is not measured, then we do not know whether the intervention has failed because it was ineffective at changing the target behaviours and therefore could not impact on the health outcomes, or if diet and physical activity behaviours were changed but the intervention was not effective at preventing adverse health outcomes. If the latter, this suggests that these behaviours are not important mechanisms of prevention, or that the magnitude of change achieved was not clinically significant even if it were statistically significant. The actions following an intervention would differ depending on our understanding of the process. For example, if the intervention was not successful at changing the target behaviour, or the magnitude of change was not large enough to have a clinical impact on disease processes, then the content could be reviewed to identify how

to improve the intensity, timing, delivery etc to achieve the required behaviour change. If the lack of effect was due to the behaviours being unimportant mechanistically then we could make recommendations that future interventions need to target different mechanisms. Alternatively, if behaviour change interventions are demonstrated to be effective at preventing adverse health outcomes but the behaviours are not measured, then we may incorrectly assume that the mechanism was related to a change in behaviour and invest in the roll out of this type of intervention, whereas the effect may have resulted from unintended (and unmeasured) consequences of the intervention.

This systematic review of systematic reviews employed rigorous methods. We conducted extensive searches of bibliographic databases and grey literature sources, although the included reviews were only peer reviewed journal articles. All screening was carried out in duplicate and data extraction and quality assessments were validated using standardised protocols. The protocol for the review was published as a peer reviewed paper [40] and on PROSPERO (CRD42016046302). There were some deviations in methods and reporting from the published protocol. For example, the volume of data we retrieved in the 89 systematic reviews that met the wider inclusion criteria (i.e. reporting the effectiveness of behaviour change interventions delivered in pregnancy on either behavioural or health-related outcomes) was unmanageable to synthesise into one review. Therefore, we made the decision to split the work into a larger programme of reviews with three distinct aims (as described in the introduction section) which is not what we originally planned and had outlined in the published protocol [40]. Given this, it was beyond the scope of this review to examine either the effectiveness of interventions on health-related outcomes, or according to the behavioural content of the interventions reported by the systematic reviews. For example, in some smoking reviews, interventions were categorised and analysed according to whether they were 'counselling interventions', 'self-help', 'health education' etc with some observed differences in results. However, synthesising the behavioural content and features of intervention delivery (length, format, intensity, provider) across all behaviours is complex and will be the focus of a subsequent review of reviews which will address aim three in our wider research programme.

As with all systematic reviews, we were limited by the availability of data and this was a particular issue for alcohol behaviours. However, one of the aims of a systematic review of systematic reviews is to describe the current extent of evidence and the gaps to inform future research, therefore this is less of a limitation than in a systematic review of primary studies which aims to pool data. The quality of the included systematic reviews was also considered to be good, with no reviews being categorised as low quality overall. However, only about half of the included systematic reviews implemented risk of bias assessments being carried out by two researchers or assessed the likelihood of publication bias. The majority of intervention data included in the systematic reviews originated from high-income countries followed by upper-middle-income countries which is a major limitation. The results of this systematic review of systematic reviews are unlikely to be relevant to low- or lower-middle-income countries. There were some data reported in the reviews that highlights the potential differences between target behaviour change between income settings, such as the interventions aiming to increase or reduce animal-based food sources. In high-income countries, there is already over-consumption of meat-based food sources and therefore interventions are more likely to aim to reduce this consumption, whereas in low-income countries the aim to increase meat-based food sources would be to achieve adequate protein and iron intake during pregnancy. The effectiveness of interventions on the target behaviours of this review are relevant for lower-income countries. There is an increasing burden of non-communicable diseases globally which accounted for 71% of deaths in 2016, with almost double the rate among adults in low- and lower-middle-income countries compared with adults in high-income countries [79].

Diet and tobacco behaviours are the first and third top risk factors respectively for global causes of death [80]; therefore, interventions targeting these behaviours are essential, especially given the lack of generalisability between the evidence-base of interventions set in high-income countries to lower-income countries.

This systematic review and meta-analysis has identified the extent of current research across four behavioural domains, and importantly, the gaps in the evidence base which can inform future research activities. The most promising data for behaviour change in pregnancy relates to dietary behaviours. Further research required to advance this field includes measuring diet and/or physical activity behaviour change which is crucial to enhance our understanding on how to capitalise on the unique opportunity that pregnancy present for behaviour change interventions, both for public health and clinical health, and to reduce research waste. Further, due to inconsistency in the way that diet, and to some extent, physical activity data are reported and limitations in the ability to pool data for multiple behavioural outcomes, more standardised measurements should be used to facilitate future meta-analyses. Given the comparatively limited data available for alcohol interventions, and the potential for severe consequences of alcohol consumption during pregnancy such as fetal alcohol syndrome, further research should be directed to this field ensuring we build on the knowledge gained, and limitations, of existing research to date. Finally, given the importance of the double burden of disease in low- and middle-income countries, there is a clear gap in the evidence-base relating to behavioural interventions in this context.

## Supporting information

**S1 Table. PRISMA checklist.**
(DOC)

**S2 Table. Search terms.**
(DOCX)

**S3 Table. Screening tool based on the inclusion criteria of this umbrella review.**
(DOCX)

**S4 Table. JBI critical appraisal checklist for systematic reviews and research syntheses (amended).**
(DOCX)

**S5 Table. Description of included systematic reviews.**
(DOCX)

**S6 Table. Critical appraisal results for each behavioural domain.**
(DOCX)

**S7 Table. Overlap of included studies in the systematic reviews.**
(DOCX)

**S8 Table. Smoking behaviour summary of evidence from systematic reviews incorporating meta-analysis data.**
(DOCX)

**S9 Table. Smoking behaviour summary of evidence from systematic reviews reporting narrative synthesis data.**
(DOCX)

**S10 Table. Alcohol behaviour summary of evidence from systematic reviews reporting narrative synthesis data.**
(DOCX)

**S11 Table. Diet behaviour summary of evidence from systematic reviews incorporating meta-analysis data.**
(DOCX)

**S12 Table. Diet behaviour summary of evidence from systematic reviews reporting narrative synthesis data.**
(DOCX)

**S13 Table. Physical activity behaviour summary of evidence from systematic reviews incorporating meta-analysis data.**
(DOCX)

**S14 Table. Physical activity behaviour summary of evidence from systematic reviews reporting narrative synthesis data.**
(DOCX)

**S15 Table. Author reported conflicts of interest.**
(DOCX)

## Acknowledgments

We would like to acknowledge colleagues who contributed to the development of the review protocol but who were not involved in carrying out the review, or who had limited involvement which does not meet authorship requirements, including Professor Janet Shucksmith, Sarah Dinsdale and Professor Louisa Ells (Teesside University). Additionally, Susan Symonds (Teesside University) assisted with some of the full text screening, Anita Tibbs and Phoebe Orangu (Newcastle University) provided some administrative support during the screening process and Julie Hogg (Teesside University) assisted with re-running the database searches for the update.

## Author Contributions

**Conceptualization:** Nicola Heslehurst, Liane Azevedo.

**Data curation:** Nicola Heslehurst, Louise Hayes, Daniel Jones, James Newham, Joan Olajide, Louise McLeman, Catherine McParlin, Caroline de Brun, Liane Azevedo.

**Formal analysis:** Nicola Heslehurst, Louise Hayes, Liane Azevedo.

**Funding acquisition:** Nicola Heslehurst, Liane Azevedo.

**Investigation:** Nicola Heslehurst, Louise Hayes, Daniel Jones, James Newham, Joan Olajide, Louise McLeman, Catherine McParlin, Caroline de Brun, Liane Azevedo.

**Methodology:** Nicola Heslehurst, James Newham, Caroline de Brun, Liane Azevedo.

**Project administration:** Nicola Heslehurst, Louise Hayes, Daniel Jones, Joan Olajide, Louise McLeman, Catherine McParlin.

**Supervision:** Nicola Heslehurst.

**Validation:** Nicola Heslehurst, Louise Hayes, Daniel Jones, Joan Olajide, Louise McLeman, Catherine McParlin, Liane Azevedo.

**Writing – original draft:** Nicola Heslehurst, Liane Azevedo.

**Writing – review & editing:** Nicola Heslehurst, Louise Hayes, Daniel Jones, James Newham, Joan Olajide, Louise McLeman, Catherine McParlin, Caroline de Brun.

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
