## [Decision Letter · Decision Letter 0]

27 Mar 2020

PONE-D-19-32026

The effectiveness of smoking cessation, alcohol reduction, diet and physical activity interventions in changing behaviours during pregnancy: a systematic review of systematic reviews

PLOS ONE

Dear Dr. Hayes,

Thank you for submitting your manuscript to PLOS ONE. After careful consideration, we feel that it has merit but does not fully meet PLOS ONE’s publication criteria as it currently stands. Therefore, we invite you to submit a revised version of the manuscript that addresses the points raised during the review process.

We would appreciate receiving your revised manuscript by April 20th, 2020. To enhance the reproducibility of your results, we recommend that if applicable you deposit your laboratory protocols in protocols.io, where a protocol can be assigned its own identifier (DOI) such that it can be cited independently in the future. For instructions see: http://journals.plos.org/plosone/s/submission-guidelines#loc-laboratory-protocols

We look forward to receiving your revised manuscript.

Kind regards,

Michele Drehmer, Ph.D

Academic Editor

PLOS ONE

Journal Requirements:

2. We note that this manuscript is a systematic review or meta-analysis; our author guidelines therefore require that you use PRISMA guidance to help improve reporting quality of this type of study. Please upload copies of the completed PRISMA checklist as Supporting Information with a file name “PRISMA checklist”.

Reviewers' comments:

Reviewer's Responses to Questions

**Comments to the Author**

1. Is the manuscript technically sound, and do the data support the conclusions?

Reviewer #1: Yes

Reviewer #2: Yes

2. Has the statistical analysis been performed appropriately and rigorously? 

Reviewer #1: N/A

Reviewer #2: N/A

3. Have the authors made all data underlying the findings in their manuscript fully available?

Reviewer #1: Yes

Reviewer #2: Yes

4. Is the manuscript presented in an intelligible fashion and written in standard English?

Reviewer #1: Yes

Reviewer #2: Yes

5. Review Comments to the Author

Reviewer #1: Line 55: “Most reviews were high quality (67%)” � cite previously how they assessed quality

Line 55-59: Overall, there was consistent evidence for improving healthy diet behaviours

related to increasing fruit and vegetable consumption and decreasing carbohydrate intake, and fairly

consistent evidence for increase in some measures of physical activity (METs and VO2 max) and for

reductions in fat intake and smoking during pregnancy. � sugestion: put the fat intake togheter to nutrition outcomes.

Why the abstract don’t present an effect estimatives?

Introduction: the two firsts phrases need references.

Line 110 -115: What is the range of prevalence of smoking and alcohol? Point the direction of difference between high and low income countries for alcohol consumition.

Line 120-121: Explain better what is light consuption of alcohon in pregnancy.

Line 129: Explain more about the perception of physical activity during pregnancy (maybe talk about barriers vs. Evidences that show benefits of physical activity in this moment).

Line 133-148: Present references for this sentences. The last phrase is very strong, I suggest a more careful phrase about the potential benefit of the review. And, about the same phrase: What outcomes for women and children are impacted?

Materials and methods:

Present the method, isn’t necessary explain what is a systematic review of systematic reviews and potencial benefits.

Did the search strategy non-publish evidence (grey literature)? If no,why? If yes, describe.

Results: Line 277: smoking, 4 alcohol and 89 diet and/or physical activity � How many only diet and only physical activity?

In the results session: present the reference when describe the result.

Congrats for show the item conflict of interest.

Reviewer #2: This systematic review of systematic reviews presents the effectiveness of interventions during pregnancy on changing women’s behaviour, such as physical activity, diet, alcohol and smoking cessation. The manuscript is a narrative syntheses and it is clearly presented. The paper is well done and the conclusions are supported by this type of review.

6. PLOS authors have the option to publish the peer review history of their article (what does this mean?). If published, this will include your full peer review and any attached files.

Reviewer #1: No

Reviewer #2: No

---

## [Author Response · Author response to Decision Letter 0]

15 Apr 2020

Editor comments: 

To enhance the reproducibility of your results, we recommend that if applicable you deposit your laboratory protocols in protocols.io, where a protocol can be assigned its own identifier (DOI) such that it can be cited independently in the future. For instructions see: http://journals.plos.org/plosone/s/submission-guidelines#loc-laboratory-protocols

Response: As reported in the manuscript the protocol for this systematic review of systematic reviews has been published: https://doi.org/10.11124/JBISRIR-2016-003162

Please ensure that your manuscript meets PLOS ONE's style requirements, including those for file naming. The PLOS ONE style templates can be found at http://www.plosone.org/attachments/PLOSOne_formatting_sample_main_body.pdf and http://www.plosone.org/attachments/PLOSOne_formatting_sample_title_authors_affiliations.pdf

Response: The PLOS ONE style templates could not be retrieved using the urls given, however, the submission guidelines provided here: https://journals.plos.org/plosone/s/submission-guidelines have been followed.

We note that this manuscript is a systematic review or meta-analysis; our author guidelines therefore require that you use PRISMA guidance to help improve reporting quality of this type of study. Please upload copies of the completed PRISMA checklist as Supporting Information with a file name “PRISMA checklist”

Response: Supplementary file 1 contains the PRISMA checklist and has been renamed “S1 PRISMA checklist” accordingly. Please note that as this is a systematic review of systematic reviews not all of the PRISMA fields are appropriate. 

 

Response to Reviewer #1:

1. Line 55: “Most reviews were high quality (67%)” -cite previously how they assessed quality.

Response: We have added the following information in the abstract: “Quality was assessed using the JBI critical appraisal tool for umbrella reviews” and made minor edits to the abstract to keep within the 300 word limit.

2. Line 55-59: Overall, there was consistent evidence for improving healthy diet behaviours related to increasing fruit and vegetable consumption and decreasing carbohydrate intake, and fairly consistent evidence for increase in some measures of physical activity (METs and VO2 max) and for reductions in fat intake and smoking during pregnancy. - suggestion: put the fat intake together to nutrition outcomes.

Response: Thank you for your suggestion, however, the reason for having fat intake separate from the other diet behaviours in the abstract is that the evidence for fat intake was ‘fairly consistent’ (in line with the evidence for METs and VO2 max) and not ‘consistent’ (as the evidence was for increased fruit and vegetable consumption and decreased carbohydrate intake). Thus, we have not made the suggested change to the abstract. 

3. Why the abstract don’t present an effect estimatives?

Response: As this is a systematic review of systematic reviews covering the impact of interventions targeting four different behaviours during pregnancy it is not feasible to present effect sizes for such a large number of outcomes within the word limit of the abstract. In addition, meta-analysis isn’t appropriate for systematic reviews of systematic reviews. We are therefore unable to present pooled effect sizes. 

4. Introduction: the two firsts phrases need references

Response: References have now been added.

5. Line 110 -115: What is the range of prevalence of smoking and alcohol? Point the direction of difference between high and low income countries for alcohol consumition.

Response: The following information has been added:

“Prevalence of smoking among pregnant women in high income countries has declined from 20 to 35% in the 1980s to between 10% and 20% in the early 2000s, with a further decline below 10% by 2010 [15].”

and

“Prevalence of alcohol use in pregnancy varies by country. It is lowest in the World Health Organisation (WHO) Eastern Mediterranean Region (EMR) (0.2%, 95% CI: 0.1–0.9) and highest in countries in the WHO European region (25.2%, (21.6 – 29.6). Prenatal alcohol exposure is associated with preterm birth, low birth weight and Fetal Alcohol Spectrum Disorders [17-19].”

6. Line 120-121: Explain better what is light consuption of alcohon in pregnancy.

Response: The following information has been added: “(usually defined as no more than 1 to 2 units, once or twice a week)”

7. Line 129: Explain more about the perception of physical activity during pregnancy (maybe talk about barriers vs. Evidences that show benefits of physical activity in this moment).

Response: A brief summary of barriers to and enablers of physical activity during pregnancy has been added.

“Existing evidence suggests that many barriers to being physically active during pregnancy exist, including lack of consistent advice, lack of societal support, physical symptoms of pregnancy and lack of opportunity[33]. Factors that motivate women to be active during pregnancy include weight control and potential for an easier labour and delivery [34].”

Reference is already made to the established benefits of physical activity during pregnancy in Line 125 (“In light of the…… potential for public health gain through intervention, there are national and international guidelines for diet and physical activity behaviours…..”, with a reference given to the English guideline (National Institute for Health and Care Excellence. Public Health Guidelines [PH27]), which summarises the benefits of physical activity during pregnancy.

8. Line 133-148: Present references for this sentences. 

Response: References have been added in the paragraph

9. The last phrase is very strong, I suggest a more careful phrase about the potential benefit of the review. 

Response: We have rewritten this sentence and tapered the benefits of the review. 

“Such information could help to inform an interdisciplinary approach to public health around pregnancy and guide the development and delivery of cost-effective interventions with the potential to impact on short-term and long-term health outcomes for women and children.”

10. And, about the same phrase: What outcomes for women and children are impacted?

Response: We have added a sentence to indicate what the outcomes are that could be impacted upon.

“These outcomes include improved maternal weight management, reduced risk of gestational diabetes, and hypertension, lower rates of premature delivery and caesarean section and of low and high birthweight infants.”

11. Materials and methods:

Present the method, isn’t necessary explain what is a systematic review of systematic reviews and potencial benefits

Response: We thank the reviewer for their comment, but think that the systematic review of systematic reviews is a relatively unfamiliar method for many readers and that there is therefore merit in including a brief overview of the methodology. We have amended the description to make it more succinct and focused on this specific review of reviews.

“This systematic review of systematic reviews provides an overview of the existing evidence base. It compares findings of previous systematic reviews, identifies research gaps and provides direction for future research, specifically relating to intervention effectiveness across behavioural domains.” 

12. Did the search strategy non-publish evidence (grey literature)? If no,why? If yes, describe.

Response: Thank you for your comment. We had previously reported in the discussion we “conducted extensive searches of bibliographic databases and grey literature sources”. However, this information was not in the methods section. This has now been added.

“Grey literature was also included in the original search. The following databases were searched: Google Scholar, NICE Evidence search, Open Grey, The Grey literature report, National Institute for Health Research (NIHR) Journals Library, Health Technology Assessment Database, Ovid Health Management Information Centre Database, Cochrane Pregnancy and Childbirth Group. No studies were retrieved from the search of grey literature during the original search, therefore no updates of grey literature searches were performed.”

13. Results: Line 277: smoking, 4 alcohol and 89 diet and/or physical activity - How many only diet and only physical activity?

Response: The focus of this review is on only those studies that report behavioural outcomes (i.e. 4 related to alcohol, 16 to smoking and 16 to diet and/or physical activity). We have now added the details of the numbers of systematic reviews that reported diet only outcomes, physical activity only outcomes, or a combination of both diet and physical activity outcomes. 

“From these, 16 reviews reported diet and/or physical activity behaviours (18% of all diet and/or physical activity intervention reviews identified in the search).This included six reviews reporting diet outcomes only, two reviews reporting physical activity outcomes only, and six reviews reporting a combination of physical activity and diet outcomes.”

As the remaining 73 diet and/or physical activity systematic reviews did not report behavioural outcomes (and were therefore excluded from this paper) we cannot report the numbers in the same way for the diet and/or physical activity outcomes. The reviews did not consistently report the content of the interventions they included (i.e. if they included diet only components, physical activity only or a combination of both) and therefore it is not possible to comprehensively report the numbers according to the intervention content. As these additional studies were excluded from this paper anyway, we do not feel that this is a limitation in what we report for the systematic reviews reporting behavioural outcomes. Further papers from this review of review summarising the findings of systematic reviews reporting clinical outcomes (hence including the additional studies excluded from this paper) are currently in preparation. 

14. In the results session: present the reference when describe the result.

Response: References have been added as required in the results section. 

In addition, while checking our results section we noted an error in the maximum quality score awarded to the systematic reviews of alcohol interventions included in our review. This has been corrected.

---

## [Editor Report · Decision Letter 1]

22 Apr 2020

The effectiveness of smoking cessation, alcohol reduction, diet and physical activity interventions in changing behaviours during pregnancy: a systematic review of systematic reviews

PONE-D-19-32026R1

Dear Dr. Hayes,

We are pleased to inform you that your manuscript has been judged scientifically suitable for publication and will be formally accepted for publication once it complies with all outstanding technical requirements.

With kind regards,

Michele Drehmer, Ph.D

Academic Editor

PLOS ONE

---

## [Editor Report · Acceptance letter]

24 Apr 2020

PONE-D-19-32026R1 

The effectiveness of smoking cessation, alcohol reduction, diet and physical activity interventions in changing behaviours during pregnancy: a systematic review of systematic reviews 

Dear Dr. Hayes:

I am pleased to inform you that your manuscript has been deemed suitable for publication in PLOS ONE. Congratulations! Your manuscript is now with our production department. 

With kind regards,

on behalf of

Dr. Michele Drehmer 

Academic Editor

PLOS ONE